# On Associations between Fear-Induced Aggression, *Bdnf* Transcripts, and Serotonin Receptors in the Brains of Norway Rats: An Influence of Antiaggressive Drug TC-2153

**DOI:** 10.3390/ijms24020983

**Published:** 2023-01-04

**Authors:** Vitalii S. Moskaliuk, Rimma V. Kozhemyakina, Tatyana M. Khomenko, Konstantin P. Volcho, Nariman F. Salakhutdinov, Alexander V. Kulikov, Vladimir S. Naumenko, Elizabeth A. Kulikova

**Affiliations:** 1Institute of Cytology and Genetics, Siberian Branch of Russian Academy of Sciences (SB RAS), 10 Akad. Lavrentyeva Ave., 630090 Novosibirsk, Russia; 2N.N. Vorozhtsov Novosibirsk Institute of Organic Chemistry, SB RAS, 9 Akad. Lavrentieva Ave., 630090 Novosibirsk, Russia

**Keywords:** brain-derived neurotrophic factor, TC-2153, aggression, domestication, regulatory exon, serotonin receptor, brain, rat, mRNA

## Abstract

The *Bdnf* (brain-derived neurotrophic factor) gene contains eight regulatory exons (I–VIII) alternatively spliced to the protein-coding exon IX. Only exons I, II, IV, and VI are relatively well studied. The BDNF system and brain serotonergic system are tightly interconnected and associated with aggression. The benzopentathiepine TC-2153 affects both systems and exerts antiaggressive action. Our aim was to evaluate the effects of TC-2153 on the *Bdnf* exons I–IX’s expressions and serotonin receptors’ mRNA levels in the brain of rats featuring high aggression toward humans (aggressive) or its absence (tame). Aggressive and tame adult male rats were treated once with vehicle or 10 or 20 mg/kg of TC-2153. mRNA was quantified in the cortex, hippocampus, hypothalamus, and midbrain with real-time PCR. Selective breeding for high aggression or its absence affected the serotonin receptors’ and *Bdnf* exons’ transcripts differentially, depending on the genotype (strain) and brain region. TC-2153 had comprehensive effects on the *Bdnf* exons’ expressions. The main trend was downregulation in the hypothalamus and midbrain. TC-2153 increased 5-HT_1B_ receptor hypothalamusc mRNA expression. For the first time, an influence of TC-2153 on the expressions of *Bdnf* regulatory exons and the 5-HT_1B_ receptor was shown, as was an association between *Bdnf* regulatory exons and fear-induced aggression involving genetic predisposition.

## 1. Introduction

Brain-derived neurotrophic factor (BDNF) is the most abundant neurotrophin in the brain and is involved in neuro- and gliogenesis, neuroprotection, long- and short-term synaptic plasticity, and synaptogenesis (for a review, see [1]). This variety of functions is partly explained by its sophisticated gene structure. The rodent *Bdnf* gene consists of eight 5′ noncoding exons (I–VIII) alternatively spliced to a common 3′ protein-coding exon [2]. The numerous *Bdnf* RNA splice variants eventually encode the same BDNF protein. It has been shown that these eight regulatory exons undergo differential expression throughout distinct brain regions and brain developmental stages [3]. Moreover, 5′ regulatory exons serve to localize mRNAs to different parts of a neuron [4], where local translation and prompt availability of BDNF are crucial for neuroplasticity and synaptic functioning (for a review, see [5]). Disruption of untranslated exons I, II, IV, and VI diversely affects dendritic morphology and spine arborization in basal and apical dendrites [6]. Furthermore, data obtained over the last decade implicate the noncoding *Bdnf* exons in the regulation of behavior and pathological conditions. Downregulations of exons IV and VI are associated with elevated depressive-like behavior [7], while underexpression of exon VII is implicated in heightened depression and anxiety [8]. Reductions in exons II, IV, and VI’s expressions are observed in a mouse model of Huntington’s disease [9]. On the other hand, overexpression of the regulatory *Bdnf* exons is mainly associated with increased stress and fear conditioning. Enhanced transcriptions of exons I and IV in fear-conditioned animals have been documented [10,11]. Notably, distinct types of stress produce different alterations in the expressions of exons I, IV, and VI [12,13]. While single-immobilization stress leads to the downregulations of exons I and IV, these two exons are upregulated under the influence of foot shock stress [12]. At the same time, the expressions of exons IV and VI are increased after acute stress [13]. Although all these studies are mainly dedicated to exons I, II, IV, and VI, there is very little information on the cellular and behavioral roles of the other four regulatory *Bdnf* exons: III, V, VII, and VIII.

There is interest in the involvement of BDNF in the mechanism of aggressive behavior [14,15,16]. Some studies point to the elevation of aggression induced by central BDNF administration in mice [17] and a high BDNF protein level in the aggressive AB–Halle (ABH) mouse strain [15], while other research indicates an association between *BDNF* disruption and aggressiveness both in animals and humans [16,18,19]. Therefore, the participation of BDNF in the mechanism of aggression is controversial and may depend on the 5′ regulatory exons. At present, there is only one study showing associations between the different regulatory *Bdnf* exons’ expressions and aggression [20]. That report revealed that selective disruption of exons I and II, but not exons IV and VI, results in elevated offensive territorial aggression in mice [20]. These types of depletion are accompanied by changes in the serotonergic system: mutations of exons I and II are associated with an increase in serotonin transporter and 5-HT_2A_ receptor gene expressions [20]. The connection between aggression and the serotonergic system is well known, and serotonin is considered one of the main neurotransmitters regulating this complex type of behavior (for a review, see [21]). Genetic variants of the serotonin receptors 5-HT_1A_, 5-HT_1B_, 5-HT_2A_, and 5-HT_7_ [22,23,24,25] are implicated in this type of behavior. Moreover, serotonin receptors and BDNF share a reciprocal modulating link: BDNF injection raises the expressions of genes encoding the 5-HT_1A_, 5-HT_1B_, and 5-HT_2A_ receptors [26,27], whereas activation of the 5-HT_2A_ receptor in turn regulates *Bdnf* gene expression [28,29,30].

On the other hand, the molecular mechanisms underlying aggression involve multiple brain systems and are associated with environmental as well as hereditary factors. Two strains of wild Norway rats selectively bred for 90 generations at the Institute of Cytology and Genetics (Novosibirsk, Russia) for enhancement (aggressive rats) or absence (tame rats) of aggression toward humans are a suitable model for investigating mechanisms of fear-induced aggression and domestication. These animals differ in a number of behavioral characteristics: the aggressive rats manifest greater fear [31,32], high anxiety [33], and a reduction in learning abilities and memory compared with the tame animals [34]. Moreover, the aggressive and tame rats demonstrate significant alterations in their brain serotonergic and BDNF systems [14,31,35,36,37,38,39,40]. Nonetheless, the function of the noncoding regulatory *Bdnf* exons in the mechanisms of aggression toward humans and domestication has not been studied yet.

Recently, it was shown that the novel antidepressant benzopentathiepine TC-2153—when administered acutely or chronically—significantly diminishes the fear-induced aggression in aggressive rats and, after acute administration, has an anxiolytic effect on both tame and aggressive Norway rats [41]. TC-2153 is known to elevate the expression of the common protein-coding exon IX of the *Bdnf* gene [42] and to influence the serotonergic system in mice [43,44,45,46]. Earlier, it has been shown that the 5′ untranslated regulatory exons can be affected by the administration of classic antidepressants [7,47,48]. On the basis of these findings, we propose in this study that the new antidepressant TC-2153 can differently affect the expressions of the *Bdnf* exons, and these results can shed light on the psychotropic effects of TC-2153.

Accordingly, the aim of this study was to investigate the expressions of all the *Bdnf* exons (I–IX) and serotonin receptors 5-HT_1A_, 5-HT_1B_, 5-HT_2A_, and 5-HT_7_ (encoded by genes *Htr1a*, *Htr1b*, *Htr2a*, and *Htr7*, respectively) in the model of genetically determined fear-induced aggression (or its absence) after acute administration of the novel antidepressant TC-2153.

In this work, for the first time, we showed that TC-2153 affects the regulatory *Bdnf* exons depending on the genotype (strain) and brain region and upregulates *Htr1b* mRNA. The regulatory *Bdnf* exons I–VIII were found to be associated with fear-induced aggression involving genetic predisposition.

## 2. Results

### 2.1. Effects of Acute TC-2153 Treatment on mRNA Levels of Bdnf Untranslated Exons I–VIII and Coding Exon IX in the Cortex of Aggressive and Tame Rats

The two-way analysis of variance (ANOVA) revealed significant effects of the genotype factor on the expressions of exons I–III, V, VII, and VIII. At the same time, no effect of the drug factor or of the interaction of the two factors on the expressions of exons I–IX was detected (Appendix A). Aggressive rats manifested higher expressions of exons I, II, III, V, VII, and VIII compared with the tame animals (Figure 1).

### 2.2. Effects of Acute TC-2153 Treatment on mRNA Levels of Bdnf Untranslated Exons I–VIII and Coding Exon IX in the Hippocampus of Aggressive and Tame Rats

We found significant effects of the genotype factor on the expressions of all 5′ noncoding exons and of the drug factor on exons V and VI, while there were no effects of the factors’ interaction on the expressions of the studied exons (Appendix A). Expression levels of exons II–VIII were lower in the hippocampus of aggressive rats compared with tame ones, while expression of exon I proved to be elevated in aggressive rats. TC-2153 at the dose of 10 mg/kg increased the mRNA level of exon VI. At the same time, the post hoc analysis of exon V expression did not reveal any differences between the groups (Figure 2).

### 2.3. Effects of Acute TC-2153 Treatment on mRNA Levels of Bdnf Untranslated Exons I–VIII and Coding Exon IX in the Hypothalamus of Aggressive and Tame Rats

In the hypothalamus, the two-way ANOVA uncovered significant effects of the genotype on the expressions of common exon IX and exon I and of the drug factor on the expressions of exons I, V, and VII, while no effect of a genotype × drug interaction was observed (Appendix A). Aggressive animals featured higher expressions of the common exon IX and of exon I than tame rats did. TC-2153 at the dose 20 mg/kg reduced the expression of exon I in both strains and the expression levels of exons V and VII in tame rats (Figure 3).

### 2.4. Effects of Acute TC-2153 Treatment on mRNA Levels of Bdnf Untranslated Exons I–VIII and Coding Exon IX in the Midbrain of Aggressive and Tame Rats

In the midbrain, with the two-way ANOVA, we found significant effects of the genotype on the expressions of exons I–V and effects of the drug factor on exons II, IV, V, VII, and VIII and common exon IX, while no effects of a genotype × drug interaction were seen (Appendix A). The expressions of exons I–V were higher in aggressive rats than in tame ones. TC-2153 significantly reduced exons II and VIII’s mRNA levels in tame rats, while the expressions of exons IV, V, and VII were significantly diminished in both strains. TC-2153 at the dose of 20 mg/kg also reduced the expression of the common exon IX in tame rats (Figure 4).

### 2.5. The Influence of Acute TC-2153 Treatment on the Htr1a mRNA Levels in Brain Structures of Aggressive and Tame Rats

We found a significant effect of the genotype on the 5-HT_1A_ receptor mRNA levels in the hippocampus (F_1,33_ = 9.42, *p* < 0.01) and midbrain (F_1,32_ = 5.69, *p* < 0.05): aggressive rats had lower *Htr1a* gene expression than the tame animals did. Nonetheless, no genotype effect was detectable in the cortex (F_1,34_ < 1) and hypothalamus (F_1,36_ = 2.06, *p* > 0.05). Acute TC-2153 administration had an impact on *Htr1a* mRNA expression in the hippocampus (F_2,33_ = 5.05, *p* < 0.05); however, the post hoc analysis did not show any difference between the groups. In other brain structures, no effect of the drug was found (cortex F_2,34_ < 1; hypothalamus F_2,36_ < 1; and midbrain F_2,32_ = 1.39, *p* > 0.05). Genotype × drug interactions were insignificant (cortex F_2,34_ < 1; hippocampus F_2,33_ < 1; hypothalamus F_2,36_ = 1.41, *p* > 0.05; and midbrain F_2,32_ = 1.02, *p* > 0.05) (Figure 5).

### 2.6. Effects of Acute TC-2153 Treatment on Htr7 mRNA Levels in Brain Structures of Aggressive and Tame Rats

We detected a higher level of 5-HT_7_ receptor mRNA in the cortex (genotype effect F_1,32_ = 15.79, *p* < 0.001) and hippocampus (genotype effect F_1,36_ = 4.45, *p* < 0.05) of aggressive rats compared with tame ones. No genotype effect was found in the hypothalamus (F_1,35_ = 1.11, *p* > 0.05) and midbrain (F_1,36_ = 1.52, *p* > 0.05). TC-2153 exerted no effect in any of the tested brain structures (cortex F_2,32_ < 1; hippocampus F_2,36_ < 1; hypothalamus F_2,35_ < 1; and midbrain F_2,36_ < 1). Genotype × drug interactions were insignificant (cortex F_2,32_ < 1; hippocampus F_2,36_ = 2.57, *p* > 0.05; hypothalamus F_2,35_ < 1; and midbrain F_2,36_ < 1) (Figure 6).

### 2.7. Effects of Acute TC-2153 Treatment on Htr1b mRNA Levels in Brain Structures of Aggressive and Tame Rats

Aggressive rats had an elevated expression of the 5-HT_1B_ receptor gene in the frontal cortex (genotype effect F_1,31_ = 6.94, *p* < 0.05) and low expression in the midbrain (genotype effect F_1,35_ = 6.79, *p* < 0.05). No genotype effect was detectable in the hippocampus (F_1,35_ < 1) and hypothalamus (F_1,35_ < 1). Acute TC-2153 administration upregulated the mRNA of this receptor in the hypothalamus of both strains of rats (F_2,35_ = 4.25, *p* < 0.05). The drug did not affect 5-HT_1B_ receptor expression in the cortex (F_2,31_ < 1), hippocampus (F_2,35_ < 1), and midbrain (F_2,35_ = 1.27, *p* > 0.05). At the same time, there were no effects of genotype × drug interaction (cortex F_2,31_ < 1; hippocampus F_2,35_ < 1; hypothalamus F_2,35_ = 1.49, *p* > 0.05; and midbrain F_2,35_ = 1.55, *p* > 0.05) (Figure 7).

### 2.8. Effects of Acute TC-2153 Treatment on Htr2a mRNA Levels in Brain Structures of Aggressive and Tame Rats

The investigation of 5-HT_2A_ receptor expression revealed a genotype effect in all the tested brain structures, except for in the cortex (cortex F_1,35_ = 3.12, *p* > 0.05; hippocampus F_1,35_ = 17.90, *p* < 0.001; hypothalamus F_1,35_ = 4.73, *p* < 0.05; and midbrain F_1,36_ = 5.06, *p* < 0.05): aggressive rats had higher mRNA levels relative to the tame animals. At the same time, no influence of the drug (cortex F_2,35_ < 1; hippocampus F_2,35_ = 1.95, *p* > 0.05; hypothalamus F_2,35_ = 2.53, *p* > 0.05; and midbrain F_2,36_ < 1) or of a genotype × drug interaction (cortex F_2,35_ < 1; hippocampus F_2,35_ < 1; hypothalamus F_2,35_ = 1.82, *p* > 0.05; and midbrain F_2,36_ < 1) was registered in any of the analyzed brain structures (Figure 8).

## 3. Discussion

In this study, primers for all eight 5′ untranslated exons of the rat *Bdnf* gene were designed, and the mRNA levels of these regulatory exons were assessed in the cortex, hippocampus, hypothalamus, and midbrain of Norway rats (which had a genetic predisposition to fear-induced aggression or its absence) after acute treatment with the novel antidepressant TC-2153.

We obtained quite diverse results. Compared with the tame rats, the aggressive rats featured elevated expression levels of exons I, II, III, V, VII, and VIII in the cortex; exons I and IX in the hypothalamus; and exons I, II, III, IV, and V in the midbrain, and underexpression of all the regulatory exons in the hippocampus except for exon I, which was upregulated (Figure 9). The genotype effects in the cortex, hypothalamus, and midbrain are in agreement with multiple research articles that indicate an association between aggression and elevated levels of the BDNF protein and its isoforms [15,49]. It should be noted that earlier, we revealed a difference between the same rat strains in the expression of protein-coding exon IX in the cortex [14]. These results failed to be replicated in the current study. As opposed to the current study, in the previous work, the animals received neither drug nor vehicle administration. The differences in the experimental design could explain the discrepancies between our present results and previously published data.

In the hippocampus, the expressions of almost all the 5′ regulatory were higher in the tame rats. At the same time, in this project, we detected no difference between the rat strains in the expression of protein-coding exon IX in the hippocampus, in good agreement with the previously obtained data [14]. It is well known that the hippocampus plays a crucial part in memory, learning, and neural plasticity. Previously, it has been reported that aggressive rats show impaired learning abilities in the Morris water maze test [34]. These disturbances in the spatial memory of aggressive rats may be explained by the reduction in the expression of *Bdnf* regulatory exons in the hippocampus revealed in the current paper.

To date, the association between *Bdnf* regulatory exons and aggression has been observed only in one paper (by Maynard and colleagues), who investigated whether a mutation of exon I, II, IV, or VI leads to territorially induced intermale aggression [20]. They showed that the lack of exon I or II aggravates such behavior, while disruption of exons IV or VI does not have such an effect. The other four exons were not tested in that study. At first glance, our results contradict these findings: exon I expression turned out to be elevated in the cortex, hippocampus, hypothalamus, and midbrain of aggressive rats, exon II was upregulated in the cortex and midbrain and underexpressed in the hippocampus, exon IV was also upregulated in the midbrain and downregulated in the hippocampus, while the expression of exon VI was lower in the hippocampus of aggressive rats. Nonetheless, it should be mentioned that territorial intermale aggression is not the same type of aggressive behavior as fear-induced aggression, and they are probably regulated by different mechanisms. Moreover, it has been reported that a knockout of *Bdnf* causes aggressive behavior [16,50,51]. On the contrary, genetic and ethological models of aggression indicate a positive association between BDNF level and aggressive behavior [14,15,17]. Thus, our study confirmed the involvement of exons I and II in aggressive behavior, as documented by Maynard and colleagues. On the other hand, our findings indicate the participations of exons IV and VI as well, which may be explained by the difference in the type of aggression. Moreover, in this study we, for the first time, detected changes in the expression of the other four regulatory exons—III, V, VII, and VIII—in the brains of aggressive rats in the model of fear-induced aggression compared with tame animals. Overall, the patterns of the 5′ exons’ expressions depend on the brain region.

We detected a higher mRNA level for the majority of the regulatory *Bdnf* exons in the midbrain of aggressive rats compared with tame ones. It is well known that the cell bodies of 5-HT (serotonin) neurons are located in the midbrain raphe nuclei [52], ensuring neurotransmitter synthesis. The brain 5-HT system is known to play a central role in mechanisms of aggression [21]. On the other hand, BDNF takes part in the development of serotonin neurons at the early stages of ontogenesis of the central nervous system [53,54]. In a mature brain, these two systems are interdependent and regulate each other (for a review, see [55]). Nevertheless, it is still unclear what function the regulatory *Bdnf* exons perform in the interaction between the BDNF system and the 5-HT system and in the involvement of this crosstalk in the mechanisms of genetically determined aggressive behavior. Pharmacological and genetic studies suggest that the serotonin receptors 5-HT_1A_, 5-HT_7_, 5-HT_1B_, and 5-HT_2A_ (encoded by genes *Htr1a*, *Htr7*, *Htr1b*, and *Htr2a*, respectively) participate in the regulation of different types of aggression [22,23,24,25]. Therefore, to research the potential connection between the regulatory *Bdnf* exons and the 5-HT system, we measured the mRNA levels of these 5-HT receptors in the brains of rats with a genetic predisposition to fear-induced aggression or its absence.

We noticed differences in the expressions of 5-HT receptors between the tested rat strains. Aggressive rats were found to have lower *Htr1a* mRNA levels in the hippocampus and midbrain relative to tame rats. It is known that 5-HT_1A_ receptor agonists exert an antiaggressive effect [38,56]. Our findings suggest that a higher expression of this receptor’s mRNA can also be associated with the absence of aggression in tame animals. Moreover, in a previous report, we noted that aggressive rats have higher anxiety than tame rats in the elevated plus maze test [41]. On the other hand, a knockout of the *Htr1a* gene also aggravates anxious behavior [57]. Consequently, the revealed difference in 5-HT_1A_ receptor expression between these rat strains can offer an explanation for the previous findings.

It is well established that 5-HT_1A_ receptors can form heterodimeric complexes with 5-HT_7_ receptors [58] and thereby can regulate each other’s activity. There are contradictory data about the link between 5-HT_7_ receptors and anxiety [59,60]. At the same time, it has been previously shown that aggressive rats have a high *Htr7* mRNA level in the frontal cortex and a reduced level in the midbrain and hypothalamus [22]. In the present study, we documented an elevated expression of this receptor in the frontal cortex of aggressive rats, and (in contrast to previous data) in the hippocampus, while no difference was observed in the midbrain and hypothalamus. 5-HT_7_ receptors are reported to be involved in the regulation of memory [61]. Perhaps the elevated expressions of these receptors in the hippocampus and cortex are related to memory consolidation. At the same time, the current study provides additional evidence of the alteration of 5-HT_7_ receptor expression in the brains of rats with genetically predisposed fear-induced aggression compared with tame ones.

The observed effect of the genotype (strain) on the 5-HT_1B_ receptor was dependent on the brain region. Although in the frontal cortex, aggressive rats had a higher *Htr1b* mRNA level compared with the tame animals, and in the midbrain, the pattern was the opposite. It is well known that 5-HT_1B_ receptor agonists tend to inhibit aggressive behavior [62], whereas a knockout of this gene leads to more profound aggression [63]. Therefore, the decrease in *Htr1b* expression in the midbrain, where cell bodies of serotonergic neurons are located, may be linked with the aggression enhancement, whereas the *Htr1b* upregulation in the cortex probably represents a compensatory mechanism.

At the same time, in aggressive rats, *Htr2a* mRNA levels proved to be elevated in the hippocampus, hypothalamus, and midbrain, pointing to the importance of this gene during the selection for high aggression toward humans or its absence. The 5-HT_2A_ receptor is known to partake in other types of aggression as well, although its action is ambiguous: agonists [64], as well as antagonists [65], show an antiaggressive effect. In the present work, we demonstrated that 5-HT_2A_ receptor expression is elevated in the brains of rats with increased fear-induced aggression. Furthermore, our data indicate a possible association between 5-HT_2A_ receptor expression and *Bdnf* exon I mRNA levels. Both were elevated in the brains of aggressive rats compared with tame ones in almost all the tested brain structures. By contrast, Maynard and colleagues have previously observed an opposite association between these two genes: *Bdnf* exon I knockouts yield 5-HT_2A_ receptor overexpression in the prefrontal cortex [20]. Nevertheless, based on the current results, it is possible to hypothesize that exon I may play a part in the regulation of 5-HT_2A_ receptor expression. As described above, aggressive rats manifested a lower expression of the 5-HT_1A_ receptor in the midbrain, where it was mainly expressed presynaptically, and its activation leads to the hyperpolarization of serotonergic neurons and inhibition of serotonin signaling (for a review, see [66]). On the other hand, 5-HT_2A_ receptors are known activators of the 5-HT system (for a review, see [67]), and thus, via such expression patterns of these two receptor types, aggressive rats possess a more activated 5-HT system than tame rats do. This finding is consistent with earlier data suggesting that aggressive rats have a higher level of 5-HT in the cingulate cortex, nucleus accumbens, and putamen [33] compared with tame ones.

Furthermore, in this paper, we present an effect of acute administration of the benzopentathiepine TC-2153 on the expressions of the *Bdnf* 5′ regulatory exons (Figure 10). This influence is differentially exerted among distinct brain structures and depends on the genotype. The main impact of acute TC-2153 administration was seen in the hypothalamus and midbrain. The drug downregulated the *Bdnf* exons’ expressions in these brain structures. In tame rats, TC-2153 lowered the expressions of exons I, V, and VII in the hypothalamus and of exons II, IV, V, VII, VIII, and IX in the midbrain. In addition, in response to TC-2153 administration, aggressive rats showed a reduction in exon I expression in the hypothalamus and in the expressions of exons IV, V, and VII in the midbrain. Taking into account the results on the antiaggressive effect of TC-2153 obtained earlier [41], it is not surprising to see the main effects of this drug in the hypothalamus and midbrain because these brain structures are deeply involved in the regulation of aggressive behavior [68]. It is noteworthy that TC-2153 at the dose of 10 mg/kg elevated the expression of exon VI in the hippocampus. Therefore, TC-2153 decreased the expression of various *Bdnf* exons in the brain, except for exon VI in the hippocampus. Taken together, all our data suggest that TC-2153 is the first known drug to have such a comprehensive effect on *Bdnf* regulation, partly because the effects of other drugs have been investigated only on exons I, II, IV, and VI [7,47,69].

Earlier, it was shown that chronic TC-2153 administration raises the expression of the *Bdnf* protein-coding exon IX in the hippocampus of ASC mice [42]. In this study, we demonstrated that acute TC-2153 administration diminishes the expression of the protein-coding exon IX in the midbrain and hypothalamus of tame rats. These results are in good concordance with those obtained with the classic antidepressants fluoxetine, tranylcypromine, and desipramine. These drugs also cause a reduction in the expression of various *Bdnf* exons after acute administration but upregulate their expression during chronic administration [47,48]. The only known target for TC-2153 is striatal-enriched protein tyrosine phosphatase (STEP) [44,46,70], which is known to exert a downregulating effect on BDNF levels [71] through the dephosphorylation of kinases ERK1/2, which then fail to cease the phosphorylation of CREB: the main regulatory factor of *Bdnf* promoters. Via this mechanism, TC-2153, through the deactivation of STEP, should cause the upregulation of BDNF. We, however, revealed a downregulating effect on the expression of the *Bdnf* regulatory exons. These results may indicate the existence of another pharmacological target for TC-2153. Recently, it was found that fluoxetine binds directly to the receptor TrkB [72]. Taking into account the similarities between the effects of TC-2153 administration and those of fluoxetine [73,74,75], it is fair to assume that TC-2153 may also directly interact with TrkB. In this case, acute activation of this receptor may lead to a short-term compensatory decrease in *Bdnf* expression (as observed in a previous study after acute administration of fluoxetine [47,48]) and then to *Bdnf* upregulation when administered chronically. This hypothesis requires further research.

In this study, we found that TC-2153 increases the *Htr1b* mRNA level in the hypothalamus, suggesting that the acute antiaggressive effect of the drug may be mediated via regulation of this receptor’s expression. Furthermore, the 5-HT_1B_ receptor is a G_i_ protein-coupled receptor, and its activation induces a decrease in cAMP levels and then the dephosphorylation of protein kinase A and CREB [76], which regulate *Bdnf* transcription. This observation suggests that the downregulating effect of TC-2153 on the *Bdnf* exons’ expressions may be mediated by the influence of this drug on the 5-HT_1B_ receptor. By contrast, acute TC-2153 administration failed to affect the mRNA levels of *Htr1a*, *Htr7*, and *Htr2a*. Previously, it has been shown that chronic TC-2153 administration reduces the mRNA level of *Htr1a* [43] and inhibits functional activity and the protein level of the 5-HT_2A_ receptor [45]. Perhaps the changes in the expressions of the *Bdnf* regulatory exons—as seen after acute administration—lead to a more lasting effect on these serotonin receptors, which is attributed to the chronic TC-2153 administration.

## 4. Materials and Methods

### 4.1. Animals and Experimental Procedures

The experiments were performed on adult outbred male rats (*Rattus norvegicus*) (4–5 months old, weighing 300–350 g) selectively bred from wild rats for 90 generations at the Institute of Cytology and Genetics (Novosibirsk, Russia) for the absence (tame rats, *n* = 21) or enhancement (aggressive rats, *n* = 21) of fear-induced aggressive behavior toward humans [77,78].

The animals were housed in metal cages (50 × 33 × 20 cm) under standard laboratory conditions on a natural light–dark cycle with free access to water and feed in groups of four individuals. The study was conducted at the Centre for Genetic Resources of Laboratory Animals at the federal research center of the Institute of Cytology and Genetics, SB RAS (Novosibirsk, Russia) (RFMEFI62119X0023).

Forty-two rats (twenty-one aggressive and twenty-one tame) were used to evaluate the influence of acute administration of TC-2153 on the expression of different *Bdnf* transcripts and serotonin receptors. The new-generation antidepressant TC-2153 [8-(trifluoromethyl)benzo[f][1–5]pentathiepin-6-amine, N.N. Vorozhtsov Novosibirsk Institute of Organic Chemistry, Novosibirsk, Russia] was diluted in a solution of 0.05% Tween 20 and 0.05% DMSO as described before [45]. Rats were injected intraperitoneally (100 µL/100 g of body weight) with 10 or 20 mg/kg of TC-2153 or vehicle (a 0.05% solution of Tween 20 and DMSO) as a control. There were six groups of seven rats of both strains: (1) aggressive treated with vehicle, (2) aggressive treated with 10 mg/kg of TC-2153, (3) aggressive treated with 20 mg/kg of TC-2153, (4) tame treated with vehicle, (5) tame treated with 10 mg/kg of TC-2153, and (6) tame treated with 20 mg/kg of TC-2153. Five hours after a single TC-2153 or vehicle injection the rats were decapitated; their brains were rapidly removed on ice; and the cortex, hippocampus, hypothalamus, and midbrain were dissected, frozen in liquid nitrogen, and stored at −80 °C.

### 4.2. Real-Time PCR

Total RNA was extracted from the brain structures (cortex, hippocampus, hypothalamus, and midbrain) using the TRIzol Reagent (Ambion, Austin, TX, USA) according to a previously described procedure [41]. The extracted RNA was treated with RNase-free DNase (Promega, Fitchburg, WI, USA) and diluted to 0.125 µg/µL with diethyl pyrocarbonate-treated water. The obtained total RNA was subjected to complementary DNA (cDNA) synthesis with a random hexanucleotide mixture (BioLab Mix, Novosibirsk, Russia). The primers employed to estimate the numbers of *Bdnf* exons’ and serotonin receptors’ cDNA copies are listed in Table 1. SYBR Green I fluorescence detection (R-402 Master mix, Syntol, Moscow, Russia) was performed. As external standards, we used genomic DNA (0.06, 0.125, 0.25, 0.5, 1, 2, 4, 8, 16, 32, and 64 ng/µL) extracted from the rat livers [79,80,81]. Gene expression was evaluated as the number of cDNA copies per 100 copies of a housekeeping gene: DNA-dependent RNA polymerase II (*Polr2a*). To control amplification specificity, we performed a melting curve analysis at the end of each run for each pair of primers.

### 4.3. Statistical Analysis

The data from the assays of *Bdnf* exons’ and serotonin receptors’ mRNA expressions were checked for normality and equality of variance with Lilliefors’s test and Barlett’s test, respectively. These results are presented as means ± standard error of the means and were subjected to a two-way ANOVA with “genotype” (strain) and “treatment” as two independent factors that can interact. If any effect was significant, the difference between groups was assessed with a post hoc test (Fischer’s least significant difference test for pairwise comparisons). Statistical significance was set to *p* < 0.05. Dixon criteria were applied to identify outliers.

## 5. Conclusions

In conclusion, in this study, for the first time, we demonstrated that acute administration of the novel antidepressant and antiaggressive drug TC-2153 upregulates the mRNA of *Htr1b* in the hypothalamus of rats selectively bred for either high aggression toward humans or its absence. Moreover, the drug significantly diminished the expressions of several *Bdnf* regulatory exons in the hypothalamus and midbrain of aggressive and tame rats. Furthermore, we noted an association between different *Bdnf* untranslated regulatory exons and genetically determined fear-induced aggression. In the hippocampus, all *Bdnf* regulatory exons except exon I proved to be downregulated in aggressive rats, while in the frontal cortex, hypothalamus, and midbrain, the aggressive rats had elevated levels of various *Bdnf* regulatory exons compared with the tame animals. These data shed new light on the research into the role of *Bdnf* gene structure and of its regulation in mechanisms of aggression.

## Figures and Tables

**Figure 1 ijms-24-00983-f001:**
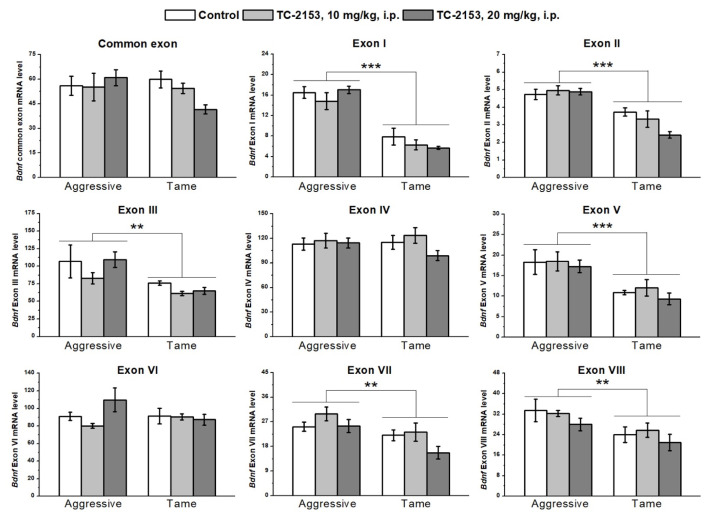
Effects of acute administration of vehicle (control) or TC-2153 at a dose of 10 or 20 mg/kg on mRNA levels of *Bdnf* (brain-derived neurotrophic factor) regulatory exons I–VIII and coding (common) exon IX in the cortex of aggressive and tame rats. The expression levels were evaluated as the number of transcript copies per 100 copies of *Polr2a* mRNA. ** *p* < 0.01 and *** *p* < 0.001 for the genotype effect (5–7 animals per group).

**Figure 2 ijms-24-00983-f002:**
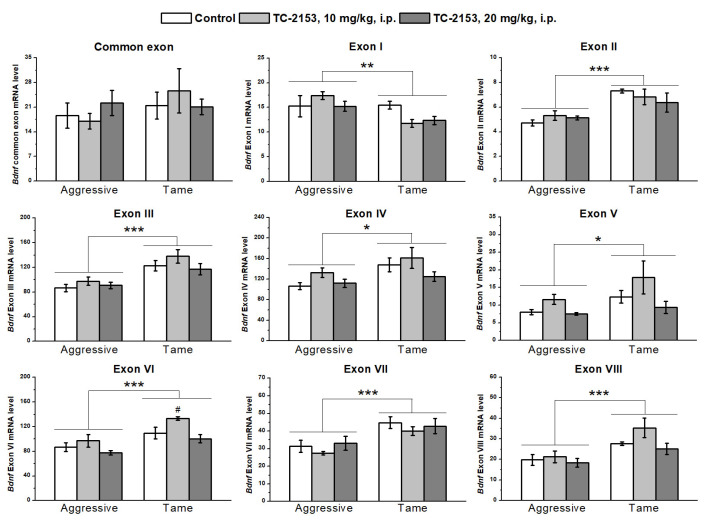
The impact of acute administration of vehicle (control) or TC-2153 at a dose of 10 or 20 mg/kg on mRNA levels of *Bdnf* (brain-derived neurotrophic factor) regulatory exons I–VIII and coding (common) exon IX in the hippocampus of aggressive and tame rats. The expression levels were evaluated as the number of transcript copies per 100 copies of *Polr2a* mRNA. * *p* < 0.05, ** *p* < 0.01, and *** *p* < 0.001 for the genotype effect; ^#^
*p* < 0.05 in comparison with the control group (5–7 animals per group).

**Figure 3 ijms-24-00983-f003:**
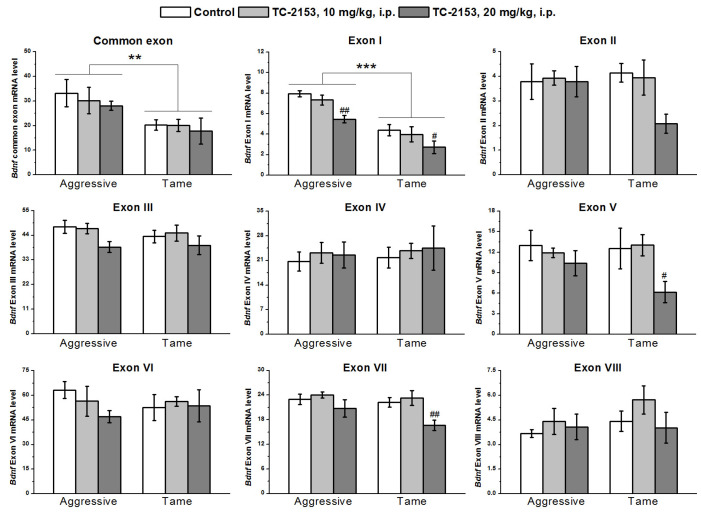
The impact of acute administration of vehicle (control) or TC-2153 at a dose of 10 or 20 mg/kg on mRNA levels of *Bdnf* (brain-derived neurotrophic factor) regulatory exons I–VIII and coding (common) exon IX in the hypothalamus of aggressive and tame rats. The expression levels were evaluated as the number of transcript copies per 100 copies of *Polr2a* mRNA. ** *p* < 0.01 and *** *p* < 0.001 for the genotype effect; ^#^
*p* < 0.05 and ^##^
*p* < 0.01 in comparison with the control group (4–7 animals per group).

**Figure 4 ijms-24-00983-f004:**
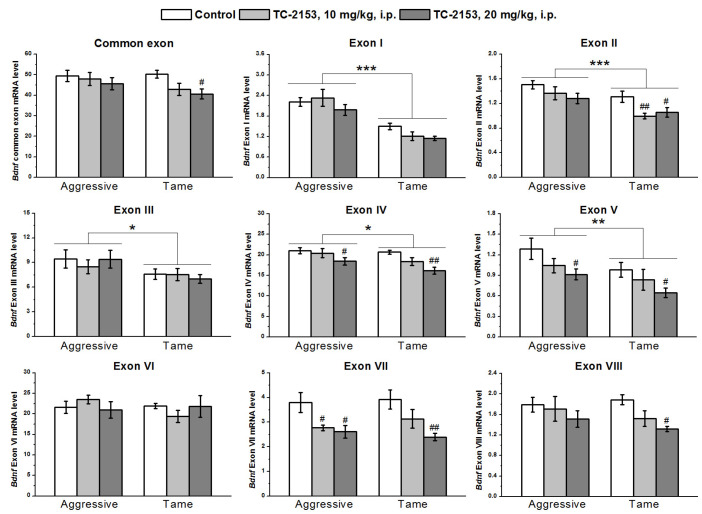
Effects of acute administration of vehicle (control) or TC-2153 at a dose of 10 or 20 mg/kg on mRNA levels of *Bdnf* (brain-derived neurotrophic factor) regulatory exons I–VIII and coding (common) exon IX in the midbrain of aggressive and tame rats. The expression levels were evaluated as the number of transcript copies per 100 copies of *Polr2a* mRNA. * *p* < 0.05, ** *p* < 0.01, and *** *p* < 0.001 for the genotype effect; ^#^
*p* < 0.05 and ^##^
*p* < 0.01 in comparison with the control group (5–7 animals per group).

**Figure 5 ijms-24-00983-f005:**
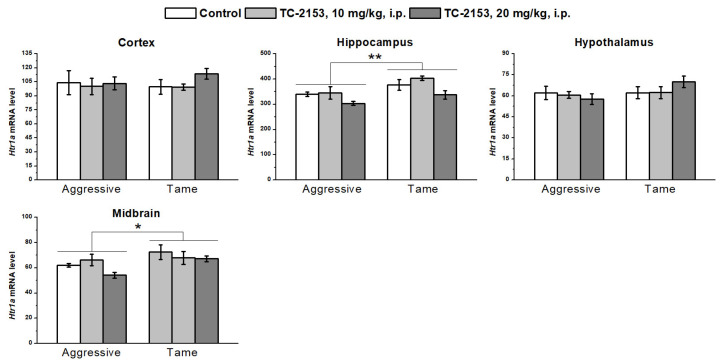
Effects of acute administration of vehicle (control) or TC-2153 at a dose of 10 or 20 mg/kg on 5-HT_1A_ receptor (*Htr1a*) mRNA levels in the cortex, hippocampus, hypothalamus, and midbrain of aggressive and tame rats. The expression levels were assessed as the number of transcript copies per 100 copies of *Polr2a* mRNA. * *p* < 0.05 and ** *p* < 0.01 for the genotype effect (5–7 animals per group).

**Figure 6 ijms-24-00983-f006:**
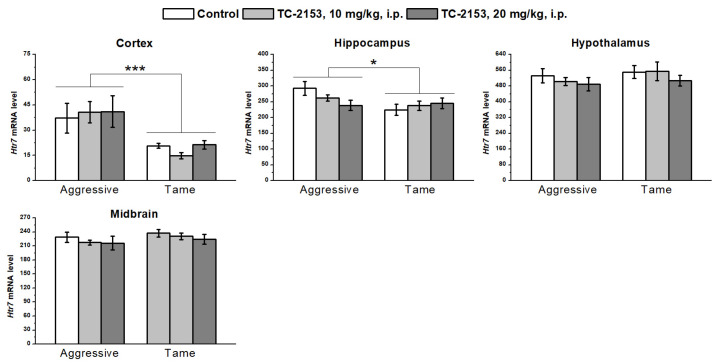
Effects of acute administration of vehicle (control) or TC-2153 at a dose of 10 or 20 mg/kg on 5-HT_7_ receptor (*Htr7*) mRNA levels in the cortex, hippocampus, hypothalamus, and midbrain of aggressive and tame rats. The expression levels were determined as the number of transcript copies per 100 copies of *Polr2a* mRNA. * *p* < 0.05 and *** *p* < 0.001 for the genotype effect (5–7 animals per group).

**Figure 7 ijms-24-00983-f007:**
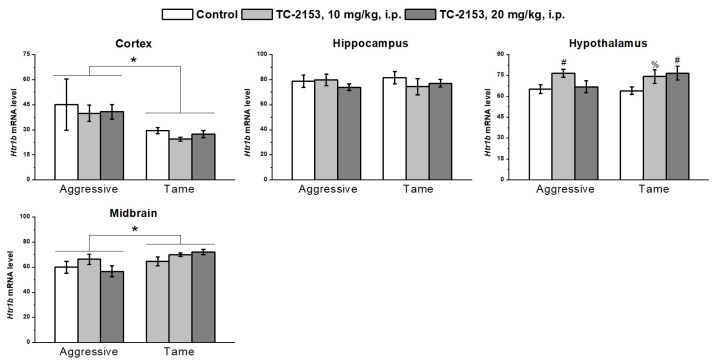
The influence of acute administration of vehicle (control) or TC-2153 at a dose of 10 or 20 mg/kg on 5-HT_1B_ receptor (*Htr1b*) mRNA levels in the cortex, hippocampus, hypothalamus, and midbrain of aggressive and tame rats. The expression levels were evaluated as the number of transcript copies per 100 copies of *Polr2a* mRNA. * *p* < 0.05 for the genotype effect; ^#^
*p* < 0.05 and ^%^
*p* = 0.062 in comparison with the control group (5–7 animals per group).

**Figure 8 ijms-24-00983-f008:**
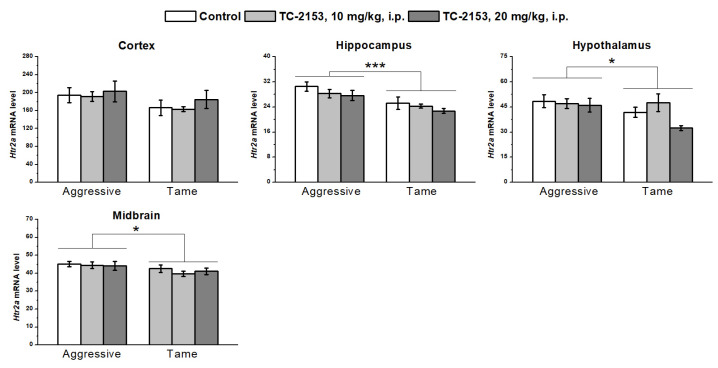
Effects of acute administration of vehicle (control) or TC-2153 at the dose 10 or 20 mg/kg on 5-HT_2A_ receptor (*Htr2a*) mRNA levels in the cortex, hippocampus, hypothalamus, and midbrain of aggressive and tame rats. The expression levels were assessed as the number of transcript copies per 100 copies of *Polr2a* mRNA. * *p* < 0.05 and *** *p* < 0.001 for the genotype effect (6–7 animals per group).

**Figure 9 ijms-24-00983-f009:**
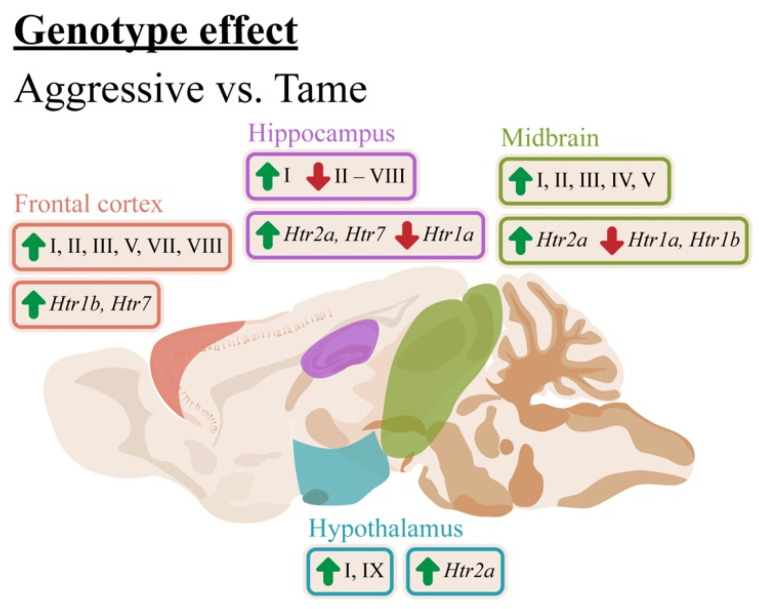
The influence of the genotype on mRNA levels of *Bdnf* (brain-derived neurotrophic factor) untranslated exons I–VIII and protein-coding exon IX and on *Htr1a*, *Htr7*, *Htr1b*, and *Htr2a* in the frontal cortex, hippocampus, hypothalamus, and midbrain of aggressive and tame rats. Green arrows indicate an elevation and red arrows indicate a reduction of expression in aggressive rats compared to the tame ones.

**Figure 10 ijms-24-00983-f010:**
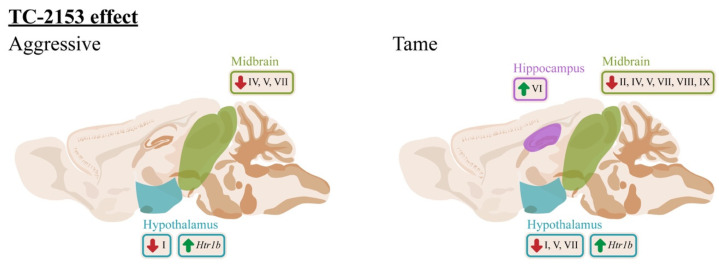
The effects of acute TC-2153 administration on mRNA levels of *Bdnf* (brain-derived neurotrophic factor) untranslated exons I–VIII and protein-coding exon IX and *Htr1a*, *Htr7*, *Htr1b*, and *Htr2a* in the frontal cortex, hippocampus, hypothalamus, and midbrain of aggressive and tame rats. Green arrows indicate an upregulation and red arrows indicate a downregulation of expression after acute administration of TC-2153.

**Table 1 ijms-24-00983-t001:** The primer sequences, annealing temperatures, and amplicon lengths.

Gene	Sequence	Annealing Temperature, °C	Product Length, bp
*Polr2a*	F: 5′-TTGTCGGGCAGCAGAACGTG-3′ R: 5′-CAATGAGACCTTCTCGTCCTCCC-3′	63	186
*Bdnf* common exon IX	F: 5′-TAGCAAAAAGAGAATTGGCTG-3′ R: 5′-TTTCTGGTCATGGATATGTCC-3′	59	257
*Bdnf* exon I	F: 5′-TGGCGACAGGGAAATCTC-3′ R: 5′-GAATGAGCGAGGTTACCAATG-3′	60	242
*Bdnf* exon II	F: 5′-CGTAAGGAAGTGGAAGAAACCGTCTA-3′ R: 5′-ATCTCAGTGTGAGCCGAACCTC-3′	65	253
*Bdnf* exon III	F: 5′-GATTCTCGCTGGATAGTTCTTTATG-3′ R: 5′-GGAGGGAAAATAGAAAGGGGTAA-3′	61	99
*Bdnf* exon IV	F: 5′-CCAGTCTCTGCCTAGATCAAATG-3′ R: 5′-GCCGATATGTACTCCTGTTCTTCAG-3′	63	121
*Bdnf* exon V	F: 5′-AAACCATAACCCCGCACACT-3′ R: 5′-CCGCACCTTCCCGCAC-3′	64	77
*Bdnf* exon VI	F: 5′-ACCAGGAGCGTGACAACAATG-3′ R: 5′-GTCCACACAAAGCTCTCGGATC-3′	65	59
*Bdnf* exon VII	F: 5′-CTTCTTACAAGTCCAAGGTCAACA-3′ R: 5′-AAGTCAAAACTTTCACTTCCTCTG-3′	64	167
*Bdnf* exon VIII	F: 5′-GGAACTTGGGATCATTCTTGTCAC-3′ R: 5′-TCTGGAGGATGCCTAAAGAGG-3′	63	117
*Htr1a*	F: 5′-CTGTCACTCTCTGCCCTACTTCTG-3′ R: 5′-CCAGAGCACATAACCCAGAGTAGT-3′	65	175
*Htr1b*	F: 5′-GTTCTTTCCCTGGTCCGCTC-3′ R: 5′-GTCCTGGTAAATGTAGTCGTCGG-3′	64	176
*Htr2a*	F: 5′-GTATATCCATGCCAATCCCAG-3′ R: 5′-CTTGATAGTCAGGAAGTAGGTG-3′	60	164
*Htr7*	F: 5′-TGATCTCGGTGTGCTTCGTC-3′ R: 5′-GTGACACTAACGAAAGGCATGAC-3′	64	115

## Data Availability

The data presented in this study are available on request from the corresponding author.

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
