# Peer review of "On Associations between Fear-Induced Aggression, Bdnf Transcripts, and Serotonin Receptors in the Brains of Norway Rats: An Influence of Antiaggressive Drug TC-2153"

_ijms, 2023, doi:10.3390/ijms24020983_

Round 1
Reviewer 1 Report
The work is well written. The introduction is informative and comprehensive and provides the correct information and references to understand the performed study. The results are clearly shown and argued in the discussion paragraph, where further evidence supporting the results is reported.
I appreciate the enrichment with tables and figures that help the readers understand the obtained data.
Reviewer 2 Report
The present work describes the effects of a single administration of a new-generation antidepressant (TC-2153, 8-(trifluoromethyl)benzo[f][1,2,3,4,5]pentathiepin-6-amine, N.N. Vorozhtsov Novosibirsk Institute of Organic Chemistry), administered to aggressive vs non-aggressive rats, in two different doses, in the expression of bdnf exons and four different 5HT receptors. Rats were killed 5 hours past TC-2153 administration, and thus the results are limited to this acute effect.
The results are clearly presented and described, but the the discussion is often too speculative.
Main concerns
Introduction:
1. When explaining how the different exons relate to behavioural changes, please clarify if these alterations are increased or decreased behaviours. Example: “overexpression of regulatory Bdnf exons is mainly associated with increased/decreased stress and fear conditioning”.
Results
1.Table 1 -4 should be moved to supplementary data, they do not add to figures 1-4.
Discussion:
L256: What is meant by “intact animals” in opposition to the current study? Please, clarify.
L264: The authors claim to have “failed” detecting differences, however these results seem to be in agreement with their previous ones. As such, saying that they have confirmed the previous results seems to be more appropriate and less confusing.
L290: The authors claim to have been the first to detect an involvement of exons II, V, VII and VIII in aggression. This is a clear overstatement, since to be able to show that the authors would have to manipulate the these exons and evaluate behaviour, which they did not, As such, all they can say is that they found differences in more aggressive rats. Please, rewrite.
L326: Again, the authors overstate that the participation of the 5-HT7 receptor in aggression. With the experiments conducted all they can say is that the receptors is altered in the aggressive rats.
L341: Here the authors state that “In the present work, we demonstrated that the 5-HT2A receptor positively correlates with aggression in the model of fear-induced aggression.”. No such correlation is presented, nor possible, since aggressive behaviour was assumed, but not measured. Please, correct.
L343: Likewise, it is not possible to say that “our data indicate a strong correlation between 5-HT2A receptor expression and Bdnf exon I mRNA levels”, correlations need to be calculated. Please correct.
L348: please replace “it is clear that exon I plays a part” by something like “ it is likely that exon I plays a part”.
